# The Feasibility and Impact of Practising Online Forest Bathing to Improve Anxiety, Rumination, Social Connection and Long-COVID Symptoms: A Pilot Study

**DOI:** 10.3390/ijerph192214905

**Published:** 2022-11-12

**Authors:** Kirsten McEwan, Harriet Collett, Jean Nairn, Jamie Bird, Mark A. Faghy, Eric Pfeifer, Jessica E. Jackson, Caroline Cook, Amanda Bond

**Affiliations:** 1College of Health, Psychology and Social Care, University of Derby, Kedleston Road, Derby DE22 1GB, UK; 2Kindred Soil, Bristol BS6 5BP, UK; 3Woodlands Breathing, Edinburgh EH27 8BW, UK; 4Biomedical Research Theme, School of Human Sciences, University of Derby, Derby DE22 1GB, UK; 5Healthy Living for Pandemic Event Protection (HL–PIVOT) Network, Chicago, IL 60612, USA; 6Catholic University of Applied Sciences, Karlstr. 63, 79104 Freiburg, Germany; 7Faculty of Psychotherapy Science, Sigmund Freud University Vienna, 1020 Vienna, Austria; 8Well with Nature, Dronfield, Derbyshire S18 4AX, UK; 9Wild Edgewalker Forest Therapy, Jersey JE3 8AF, UK

**Keywords:** anxiety, Long-COVID, COVID-19, forest bathing, health, social connection, well-being

## Abstract

Background: Long-COVID affects over 144 million people globally. In the absence of treatments, there is a need to establish the efficacy of therapies that improve patient outcomes. Forest bathing has been demonstrated to improve physical and mental outcomes but there is no evidence in Long-COVID patients. Accordingly, this pilot study sought to determine the feasibility and effectiveness of online forest bathing in adults with Long-COVID. Methods: Feasibility was assessed by monitoring retention rates and participant feedback. In a waitlist controlled, repeated measures design, 22 Long-COVID patients completed weekly online surveys during a four-week waitlist control period, before engaging in four weekly online forest bathing sessions, completing post-intervention surveys following each session. Results: In terms of retention, 27% did not provide post-intervention data, reasons for non-adherence were: feeling too ill, having medical appointments, or having career responsibilities. Compared with the waitlist control period, there were statistically significant improvements in Anxiety (49% decrease), Rumination (48% decrease), Social Connection (78% increase), and Long-COVID symptoms (22% decrease). Written qualitative comments indicated that participants experienced feelings of calm and joy, felt more connected socially and with nature, and experienced a break from the pain and rumination surrounding their illness. Conclusions: Online Forest bathing resulted in significant improvements in well-being and symptom severity and could be considered an accessible and inexpensive adjunct therapy for Long-COVID patients. Where people have limited access to in-person nature, virtual nature may offer an alternative to improve health and well-being outcomes.

## 1. Introduction

Long-COVID is a term used to describe symptoms that persist in the months and years following confirmed or probable infection with COVID-19 [1]. The clinical characterisation and treatment of Long-COVID are complex [2] because there is a wide range of symptoms (most commonly fatigue and shortness of breath) with varying levels of severity [3,4,5]. Long-COVID is a growing burden upon global healthcare services with reports suggesting that over 144 million patients are living with multi-dimensional, episodic, and disabling symptoms that broadly impact functional status and quality of life [6,7,8]. It is currently predicted that 1 in 10 COVID-19 patients will develop Long-COVID, with some Western countries suggesting this could be as many as 1 in 6 [9].

There is currently little published research on interventions for patients with Long-COVID. A review of registered trials underway [10] found 12 physical rehabilitation, 11 psychological, nine cognitive and neurorehabilitation, five pharmacological, and five natural supplement interventions for the treatment of Long-COVID. Whilst published intervention research for Long-COVID patients is emerging, Hawke [10] suggests that a wider range of intervention choices is needed for diverse Long-COVID populations. In the meantime, some researchers have offered reviews of the post-viral syndrome literature to guide intervention development. For example, Chandan [11] found that pilates, telerehabilitation, resistance exercise, and neuromodulation showed some support in symptom management. In the few published intervention studies with Long-COVID patients, physical rehabilitation has been a focus. For example, a review of physical rehabilitation [12] found improved breathing and muscle strength. In one study, digital physiotherapy improved quality of life [13], whilst another [14] found improvements in functional capacity. In a study of Hyperbaric oxygen therapy, Robbins et al. [15] found improvements in fatigue and global cognition. Whist Hawkins [16] found that aromatherapy resulted in improvements in fatigue. Self-management has been encouraged [17] and it is, therefore, important for patients to find effective evidence-based self-management interventions.

It is well-documented that exposure to nature has profound benefits for physical and mental well-being [18]. According to a review [18], exposure to nature improves cognitive function, blood pressure, physical activity, sleep, activity in the prefrontal cortex (a brain area linked to emotional regulation), and mental health. In a recent meta-analysis, Yao et al. [19] summarized that that exposure to nature decreases stress, anxiety, blood pressure, and cortisol levels, and improves health and heart-rate variability. There is evidence that during the pandemic people spent more time exposed to nature to improve well-being [20].

Going beyond mere exposure to actively engaging with nature through nature connection activities such as forest bathing, has been shown to be even more potent at improving health and well-being. Forest bathing is a slow mindful nature walk which can improve people’s physical and psychological health [21]. In Japan, it is a public health intervention available nationally on prescription which is evidenced to reduce stress [22] and improve well-being [8,23,24], blood pressure [25], immune function [26] and cardiovascular health [27].

A review detailing the correlation between nature, mindfulness, and well-being [28] suggests that during uncertain events, such as COVID-19, forest bathing may be especially important for at-risk groups, such as those experiencing depression and social isolation. In the first published study of the effects of forest bathing on mental health during the pandemic, Muro [29] found increases in positive affect, vigour, friendship, and mindfulness, and decreases in negative affect, anxiety, anger, fatigue, tension, and depressive mood. There is further evidence that during the pandemic, self-guided forest bathing reduced stress, anxiety, and depression [30]. However, these outdoor and in-person interventions are unlikely to attract patients with Long-COVID, who struggle with low energy, impaired mobility, and fear of re-infection. Indeed, patients who were shielding showed worse mental health than non-shielding participants when visiting nature during the pandemic [31]. Hence, virtual experiences of forest bathing as part of guided mindfulness activities have become more prevalent and accepted since lockdowns.

Early (pre-pandemic) studies have also found that exposure to virtual nature, such as photos, videos, nature sounds, and virtual reality can reduce stress [32], increase physiological arousal (skin conductance), and be restorative [33], as well as improving indicators of cardiovascular health and relaxation (heart rate variability) [34,35]. Helpfully, two studies directly compared exposure to real and virtual nature and found greater positive affect [33] and greater altered states of consciousness and energy [36] from exposure to real nature. Even more helpful and relevant to the current study, one study has directly compared the impact of forest bathing delivered in real settings and virtually and found that whilst both show improvements in positive affect and well-being [37], in-person forest bathing revealed larger effect sizes and improvements were more enduring.

Research undertaken during the pandemic, found that exposure to virtual nature during lockdowns (e.g., watching videos) offered a viable solution to reducing anxiety [38] and improving feelings of connectedness to the community [39]. Whilst green window views from within the home increased self-esteem, life satisfaction, and happiness and decreased depression, anxiety, and loneliness [40]. In a review of both virtual and in-person nature exposure during the pandemic, Dzhambov [41] found that exposure to nature outdoors (parks), indoors (houseplants, views), and virtually (videos of forests) were all significantly related to mental well-being, with exposure to neighbourhood greenery having the strongest effect.

Some researchers [42] have suggested that active engagement with nature through forest bathing could benefit COVID-19 patients and reduce deaths from COVID-19 [43] because of the immune-system benefits demonstrated in forest bathing research [26,44,45]. Indeed, Roviello et al. [42] suggests that exposure to phytoncides (the wood essential oils found in forests) may strengthen the immune system and reduce the severity of COVID-19 infection and found cross-sectional data indicating fewer COVID-19 deaths in more forested areas [43]. However, many Long-COVID patients struggle with low energy and impaired mobility, making it less feasible to engage with in-person nature activities such as forest bathing. White [46] suggest that virtual nature engagement could be used therapeutically with those who have mobility impairment, to improve well-being and quality of life outcomes. To date there have been no studies assessing the impact of virtual nature on patients with Long-COVID who experience mobility impairment. Therefore, this study assessed the cumulative effects of four sessions of online forest bathing on Long-COVID symptoms and psychosocial variables such as anxiety, rumination, and social connection.

## 2. Aims

The study aimed to assess whether forest bathing is feasible for adults with Long-COVID to practice online. A secondary aim was to collect pilot data assessing whether online forest bathing can improve well-being and reduce Long-COVID symptoms.

## 3. Materials and Methods

### 3.1. Design

The evaluation used a waitlist controlled, repeated measures design. A waitlist control was used because although the research team wanted to include a control group, discussions with team members who have lived-experience of Long-COVID led to the conclusion that it would not be ethical to monitor this heavily surveyed population who are rarely offered treatment, without offering an active intervention following the monitoring period. Participants completed baseline surveys for four weeks during a waitlist control period (*N* = 22 completed surveys), before engaging in four weekly online forest bathing sessions and then completing surveys post-intervention after each session (*N* = 16 completed surveys). Through the repeated-measures design, participants acted as their own controls, and their waitlist controlled and post-intervention data were matched through a participant-generated ID code. The researchers attempted to collect follow-up data one month later (*N* = 6 completed surveys).

### 3.2. Participants

Initially 26 females expressed an interest in participation; of these 22 females, White British participants aged 24–61 years old (*M* = 37.13, *SD* = 9.44 years), consented and completed surveys during the waitlist control period. Sixteen participants completed surveys post-intervention and six participants completed surveys at one-month follow-up.

### 3.3. Procedure

The forest bathing guides and researchers approached participants through social media adverts. The social media adverts featured a link to a survey on Qualtrics which featured the information sheet, consent form and surveys. Participants consented to the study and to being contacted by a forest bathing guide. Participants then completed weekly online surveys in the four weeks before the intervention started as part of the waitlist control condition. The forest bathing guides then emailed the participants to share EventBrite links to join their weekly online forest bathing sessions. After each online session, participants were emailed a link to complete their weekly post-intervention surveys. Participants were contacted again at one-month follow-up to complete the same surveys plus an additional question asking whether they had continued to practice forest bathing. A flow diagram showing this procedure and the number of participants at each stage is given in Figure 1.

### 3.4. Outcome Measures

#### Feasibility

Feasibility was assessed by monitoring retention rates from consent and waitlist control survey completion, through to post-intervention survey completion. Brief written feedback about the sessions was invited at the end of every survey. Practitioners also noted reasons for absence from online forest bathing sessions (where information was offered) and asked for general feedback at the end of the final session about any barriers to attendance and engagement.

### 3.5. Survey

The online survey comprised a total of 26 items with Likert responses which took an average of 3 min to complete. Measures included: Gender, Age, Ethnicity, Anxiety (Tension subscale of the Profiles of Mood States-POMS 6-items scored 1–5, [47]), Rumination (1 item scored 1–7, [48]), and Social Connection (Inclusion of self in others scale 1-item scored 1–7, [49]). In the absence of a concise measure of Long-COVID symptoms, the authors took the symptoms list from the WHO (World Health Organization) website (extreme tiredness; shortness of breath; chest pain or tightness; problems with memory and concentration; difficulty sleeping; heart palpitations; dizziness; pins and needles; joint pain; depression and anxiety; tinnitus, earaches; feeling sick, diarrhoea, stomach aches, loss of appetite; a high temperature, cough, headaches, sore throat, changes to sense of smell or taste; rashes) and created a Long-COVID symptom survey (15 items scored 1–7) which can be found in Appendix A. The reliability of this scale was tested using Cronbach’s alpha and found to be very reliable (α = 0.86). Only the removal of the item concerning loss of smell and taste (which showed the lowest average score) might contribute to a greater Cronbach’s alpha score (α = 0.87). The post-intervention survey also included an open-text item which invited brief written feedback about the online sessions.

### 3.6. Intervention

The 1 h online forest bathing sessions were delivered by three qualified forest therapy guides. Forest therapy guides undergo extensive training (6–12 months) and deliver Forest therapy to specific client or patient groups, considering the client/patient needs in their choice of approach and activities. In this study, guides adapted the sessions to suit the needs of people with Long-COVID. To allow for differing mobility and energy levels, one Association of Nature and forest Therapy-ANFT trained forest bathing guide offered sessions which people could attend from a local green space or garden, whilst two Nadur forest bathing trained guides who themselves had Long-COVID, offered sessions which could be attended from inside the participants home. Forest bathing group sizes ranged from 3–12 participants. The local green space/garden forest bathing sessions started with an introduction to the history and purpose of forest bathing. Participants engaged in visual activities which included: noticing colours, shapes, and movement in nature. In a listening activity, participants were invited to listen to what sounds they could hear. In a smell activity, participants were invited to smell leaf-litter and soil. In a touch activity they were invited to notice the textures of tree bark. Finally, they were invited to find a ‘sit-spot’ and observe nature. Participants also engaged in at least two sharing circles throughout the session where they were invited to feedback on their experience. The intention of sharing circles is peer learning and benefiting from the experience of others. For example, one participant may notice a unique colour of leaf causing other participants to seek out the same leaf. For participants who engaged from inside their homes due to mobility and energy issues, the guides provided photographs and videos of natural scenes and guided participants to engage mindfully with a view from their window or to engage with natural objects, houseplants, and pets.

## 4. Results

### 4.1. Feasibility

Feasibility was assessed by monitoring retention rates and capturing written and verbal feedback. Following completion of surveys during the waitlist control period, 27% of participants did not complete post-intervention surveys. On average, participants attended three out of four online sessions. Three participants withdrew from the study after only providing baseline data, two participants withdrew following their first session and two withdrew after their second session. Reasons given for withdrawing (where reported) were feeling too unwell, attending many hospital appointments, caring for a relative who had become ill, and making a phased return to work which took up energy. Brief written feedback indicated some barriers identified by three participants: ‘Not having the right footwear so wet feet didn’t help! Juggling with Zoom, feed-back, using my phone with gloves etc., was a distraction’; ‘My mother was ill and hospitalised during the study and I was able to attend one session only’.

### 4.2. Impact

Paired-samples *t*-tests were conducted for all survey variables between the waitlist control period (baseline) and post-intervention. There were statistically significant improvements between waitlist control and post-intervention in Anxiety (49% decrease), Rumination (48% decrease), Social Connection (78% increase) and Long-COVID symptoms (22% decrease) see Table 1 for descriptive and *t*-test results. In terms of Long-COVID symptoms, the symptoms with the highest average ratings were tiredness, brain-fog, poor sleep quality, depression, and anxiety, consistent with previous findings concerning enduring symptoms of Long-COVID [50]. The effect size was largest for Anxiety (*d* = 1.61 [51]). At the one-month follow-up (*N* = 6) five participants continued practicing.

The mean scores were plotted across the four weeks of the waitlist phase and the four weeks post-intervention phase. Figure 2 indicates that during the waitlist phase, rumination is slowly decreasing, social connection is stable, Long-COVID symptoms and anxiety are both increasing together, but decrease together in week 4. During post-intervention, rumination and anxiety reduce, social connection increases (aside from a single decrease back to baseline during week six) and Long-COVID symptoms stay stable but increase in the last week. All scores start to migrate back to baseline at the one-month follow-up, however, most variables are still improved compared to baseline except Long-COVID symptoms which become worse than baseline.

### 4.3. Qualitative Results

Two reviewers (one male with qualifications in art therapy and one female who has lived experience of Long-COVID, and delivers an online support forum) identified the most recurring themes to emerge within participants written qualitative feedback (42 comments, 1077 words) provided after each forest bathing session. The identified themes were divided into Major themes (mentioned > 5 times) and Minor themes (mentioned < 5 times) (see Table 2).

Feelings of calm appeared frequently in the responses given, for example, ‘I really enjoyed it, very calming, I felt less anxious and breathless afterwards. I had a call with my respiratory physician immediately afterwards and he remarked that I had less air hunger than before which may be from being calmer’ and ‘I am better because of the last hour. I don’t want to move to disturb the feeling of peace!’. This indicated that participation in virtual forest bathing can engender some of the same feelings that appear when people are physically able to be outdoors. Further studies would be needed to identify how strongly these positive psychological feelings tally with the physiological signs already observed when people are physically present [25,26,27,44]. There was also the appearance of joy and wonder within the responses, for example, ‘I really enjoyed being outside. I was only 5 min from home but felt invigorated’. This suggests that even online forest bathing can provide a strong sense of awe and curiosity about nature.

The positive statements made about connecting with nature, mirror the positive statements made about connecting to other people, for example, *‘It’s life-changing. Being able to actively feel like I’m connected to nature whilst being bedridden has absolutely been life- changing. It created a mini community where we have been able to support each other through really difficult times and has honestly helped more than anything else I’ve done in the last two years’*. Being able to be with other people, in the context of sharing a virtual space centred on a common appreciation of the natural world offered a positive experience in terms of experiencing a sense of community and support.

Participation enabled ‘time out’ and an escape and distraction from rumination upon symptoms; for example, ‘It was like an hours break from 16 months of pain and tiredness’ and ‘So nice to have a break from myself. I didn’t think about how ill I feel for the whole hour. Honestly so grateful for that experience’. That the sessions were regular, structured, and led by a practitioner contributed to this. Some participants have been able to take the experience of the virtual forest bathing and reapply skills and techniques learnt to their everyday lives; in particular, how they cope with chronic illness.

Less frequently occurring themes were those that identified barriers to full and active participation. Some technical difficulties were reported, as were problems with being able to be available to properly participate. Some of the latter are related to participants’ own health or that of those for whom they may have caring responsibilities. There was disappointment expressed about the sessions coming to an end, compounded when not all sessions could be attended. There was a wish expressed for the online meetings to become physical in-person meetings, although the strong feelings about the power of meeting others in whatever format remained.

Overall, the qualitative responses indicate that at the subjective level there is a real benefit to participation that comes through attention to nature and an enhanced sense of interpersonal connections with others. The shared understanding of the effects of Long-COVID and the shared appreciation of nature came together in virtual forest bathing sessions, to enable positive post-intervention effects to be felt.

## 5. Discussion

Forest bathing has been consistently shown to improve physical and mental health outcomes, however, as yet no studies have explored its impact on outcomes for Long-COVID patients. Accordingly, this pilot study sought to determine the feasibility and effectiveness of online forest bathing in adults with Long-COVID. Participants completed weekly online surveys during a four-week waitlist control period, before engaging with four weekly online forest bathing sessions and completing post-intervention surveys immediately following.

In terms of feasibility, the study had a reasonable rate of withdrawal given the health challenges faced by individuals struggling with Long-COVID. Following consent and provision of waitlist control period data, 27% did not provide post-intervention data. Only four of these participants withdrew from the study following attendance of 1–2 forest bathing sessions. Reasons for withdrawal or non-attendance at sessions were: feeling too ill, having medical appointments, having carer responsibilities, or making a phased return to work which took up energy. On average, the remaining participants attended three out of four forest bathing sessions. There are few published intervention studies focusing on Long-COVID and reporting retention rates with which to compare this retention rate. One exception was Estebanez-Pérez et al. [14] who had already allowed for a 40% non-adherence rate (i.e., adherence was classed as Long-COVID patients attending >12 out of 20 sessions of digital physiotherapy) and so achieved full adherence.

Compared with the waitlist control period, the survey responses during the intervention period revealed statistically significant improvements in Anxiety (49% decrease), Rumination (48% decrease), Social Connection (78% increase), and Long-COVID symptoms (22% decrease). These improvements offer early evidence of the effectiveness of online forest bathing in improving the well-being of people with Long-COVID and suggest a fully powered controlled trial might offer further generalisable evidence. These improvements are consistent with those seen from other virtual nature exposure activities offered to the public during the pandemic, such as Zabini et al. [38] who found reductions in anxiety and Van Houwelingen-Snippe et al. [39] who found improvements in community connectedness. The improvements are also consistent with the findings of [37], which saw improvements in well-being following virtual forest bathing. The large increase in social connection was also mirrored in the written qualitative comments provided by participants. In the written comments, several participants refer to being bedridden and having such low mobility and energy that they were unable to leave the house. It is likely therefore that social connection was an especially salient issue for this population and their low baseline scores in social connection certainly indicated this.

Written qualitative comments indicated that participants felt more socially connected, felt more connected with nature, experienced feelings of calm and joy, and experienced a break from the pain and rumination surrounding their symptoms. There is limited qualitative data from previous forest bathing studies; however, a study by Kjellgren and Buhrkall [36] captured qualitative data from participants experiences of exposure to real and virtual nature. For the real nature exposure condition they found themes of intensified sensory perception; a feeling of harmony and union with nature; well-being and quality of life; renewed energy and awakening; “here-and-now” thinking; and a sense of tranquillity. The connection with nature, in-the-moment thinking (as opposed to rumination) and low and high arousal positive effects such as feeling tranquil and energetic seen in Kjellgren and Buhrkall’s [36] data are similar to the themes of feeling more connected with nature, the break from rumination (being in the moment) and calm and joy derived from the current study. In contrast, the qualitative comments from Kjellgren and Buhrkall’s [36] virtual nature condition revealed positive emotions, but also a restlessness and anxiety; lack of concentration; a sense of being cut off from nature’s sensory input; a longing to be in ‘real’ nature. In the current study, only two participants commented on a preference for real nature.

Intervention studies for Long-COVID are still in development [10] and the few published intervention studies have so far focused on physical rehabilitation to improve breathing, muscle strength, and functional capacity [12,14], oxygen therapy to improve fatigue, and global cognition [15], and aromatherapy to improve fatigue [16]. None of these intervention studies have focused on exposure to nature or active engagement with nature through forest bathing. They have also not focused on outcomes such as anxiety, social connection or self-reported symptoms. In these respects, the current study offers a unique insight into an intervention which is feasible to adhere to and can improve these outcomes for people with Long-COVID. Offering a wider range of interventions [10] and self-management approaches has been encouraged [17] by researchers. This study offers promising early evidence for a new type of accessible intervention which can improve health, well-being, and social outcomes in Long-COVID patients.

In terms of Long-COVID symptoms, the symptoms with the highest average ratings were tiredness, brain-fog, poor sleep quality, depression, and anxiety, consistent with [50]. Given that these were clearly salient symptoms, the largest effect size was found for anxiety and large improvements in social connection reported in surveys and written feedback; future interventions should focus on improvements in fatigue, brain-fog, sleep, anxiety, and social connection in Long-COVID patients.

### Limitations and Future Directions

Although this pilot study attracted 22 participants during the waitlist control period, future studies should seek to recruit a larger sample size from support groups and medical clinics. Initially our social media posts advertised the sessions as ‘walks’ (i.e., ‘join us for an online forest bathing walk’), therefore those with low mobility and low energy felt they would not be able to participate because they expected walking to be involved. Videos of the research team explaining the research process and the online forest bathing sessions would have been helpful at the recruitment stage. One of the researchers (HC) took lots of questions over social media and subsequently produced a video. Re-infection, carer responsibilities, brain-fog, low energy, medical appointments, and return to work were all reasons for people to withdraw. Therefore, future studies should reduce the duration of the recruitment phase, offer a shorter duration of the waitlist control phase and allow participants to join other guides sessions so there is flexibility around session dates and times to reduce dropout due to illness and hospital appointments. The study sought to obtain follow-up data but experienced significant dropouts at one month follow-up (*N* = 6). Future research should aim to collect this longer-term data to assess how enduring the benefits of virtual nature are. There is some indication that benefits of virtual nature exposure are only observed in the short-term [38].

Although the duration data recorded in Qualtrics indicated that surveys took three minutes on average to complete, shorter surveys are desirable for a low energy, low mobility population who are often asked to complete medical surveys as part of their monitoring or treatment. Some participants expressed disappointment when the sessions were coming to an end and requested an increase in the number of future sessions (e.g., 12 sessions). Finally, future sessions should facilitate the continuation of social connection through the guide suggesting that participants might want to share participants’ contact details with each other.

## 6. Conclusions

In comparison with waitlist control, forest bathing was associated with statistically significant improvements in Anxiety, Rumination, Social Connection, and Long-COVID symptoms. Especially striking was the large increase in social connection following the forest bathing sessions. Written qualitative comments indicated that participants experienced feelings of calm and joy, felt more connected with other people and nature, and experienced a break from the pain and rumination surrounding their symptoms. In situations where people have limited access to in-person or ‘real’ nature engagement experiences, virtual nature may offer an alternative to connect with nature and experience its associated well-being benefits. Online forest bathing could be considered an accessible and inexpensive adjunct therapy for patients wishing to manage their condition. Limited follow-up data and comments indicated that more sessions are needed to maintain the benefits experienced during the sessions. Future research should aim to recruit a larger sample size via support-groups and medical clinics, evaluate a longer programme of sessions (e.g., 12 sessions) and compare the intervention against treatment as usual.

## Figures and Tables

**Figure 1 ijerph-19-14905-f001:**
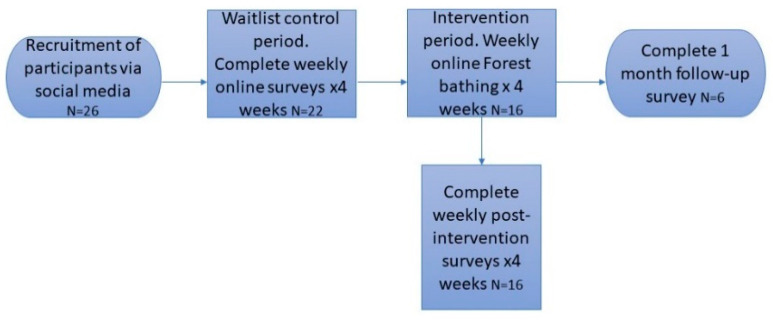
Flow diagram of the procedure for participants.

**Figure 2 ijerph-19-14905-f002:**
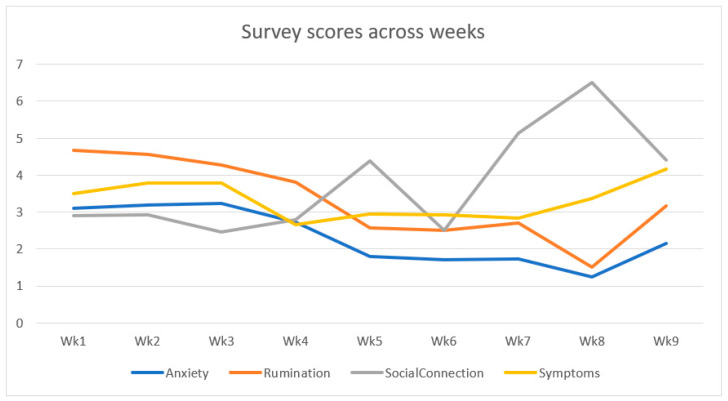
Participants’ survey scores across all weeks from the waitlist phase (weeks 1–4), post-intervention phase (weeks 4–8) to one-month follow-up (week 9).

**Table 1 ijerph-19-14905-t001:** Changes in outcome variables between waitlist period and post-intervention.

	Waitlist Control		Post-Intervention			
	*M*	*SD*	*M*	*SD*	*t*	*d*
Anxiety (POMs) ***	3.37	0.95	1.73	0.62	8.03	1.61
Rumination ***	4.79	1.41	2.50	1.10	5.84	1.19
Social Connection ***	2.68	1.11	4.76	1.51	−6.10	−1.22
Long-COVID Symptoms **	3.99	1.16	3.11	1.15	3.65	0.73

** *p* < 0.01, *** *p* < 0.001.

**Table 2 ijerph-19-14905-t002:** Major and minor themes from written qualitative data (*number of times mentioned*).

Major Themes	Minor Themes
Feelings of joy and appreciation -frequently expressed with terms like ‘enjoyed’ and *‘loved’ (15)*	Barriers to participation–technical problems, other responsibilities, limited time *(4)*
Appreciation of being able to take time out for self *(9)*	Physical benefits of participation *(2)*
Feelings of calm, peace, and comfort away from pain and rumination about symptoms *(6)*	Disappointed that sessions ended–appreciated the contact with other people *(2)*
Having a good connection to nature *(5)*	Outdoor green spaces preferred *(2)*
Having a connection to other people *(5)*	
Better resourced after participation–new skills learnt *(5)*	

## Data Availability

Anonymised data is available on request by contacting the corresponding author.

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
