# Peer review of "The Feasibility and Impact of Practising Online Forest Bathing to Improve Anxiety, Rumination, Social Connection and Long-COVID Symptoms: A Pilot Study"

_ijerph, 2022, doi:10.3390/ijerph192214905_

Round 1

Reviewer 1 Report

McEwan et al. assessed the effect of forest bathing over long-COVID patients as a pilot basis. The findings seem very interesting and important for the science of Forest Therapy and Health. I would like ask about the following very few points to improve the manuscript.

1)    The title could be shortened as, “Feasibility and impact of practising online Forest bathing to improve anxiety, rumination, social connection and Long-COVID symptoms: A pilot study

2)    I am not sure, how the researchers assessed the feasibility of practicing online forest bath. If they have explanations against it, that could be emphasized as well.

However, the manuscript has been written in an impressive manner and I appreciate authors for their work and recommend its Acceptance if addressed well with reviewers’ comments and/or suggestions.

Author Response

Reviewer 1

McEwan et al. assessed the effect of forest bathing over long-COVID patients as a pilot basis. The findings seem very interesting and important for the science of Forest Therapy and Health. I would like ask about the following very few points to improve the manuscript.

  • The title could be shortened as, “Feasibility and impact of practising online Forest bathing to improve anxiety, rumination, social connection and Long-COVID symptoms: A pilot study

Author response: Thank you, we have now changed the title to your suggestion

  • I am not sure, how the researchers assessed the feasibility of practicing online forest bath. If they have explanations against it, that could be emphasized as well.

Author response: We have now provided paragraphs in the abstract, methods, results and discussion sections which report feasibility as follows:

Abstract: In terms of retention, 27% did not provide post-intervention data, reasons for non-attendance were: feeling too ill, having medical appointments, or having carer responsibilities.

Methods: Feasibility was assessed by monitoring retention rates from consent and waitlist control survey completion, through to post-intervention survey completion. Brief written feedback about the sessions was invited at the end of every survey. Practitioners also noted reasons for absence from online forest bathing sessions (where information was offered) and asked for general feedback at the end of the final session about any barriers to attendance and engagement.

Results: Feasibility was assessed by monitoring retention rates and capturing written and verbal feedback. Following completion of surveys during the waitlist control period, 27% of participants did not complete post-intervention surveys. On average participants attended three out of four online sessions. Three participants withdrew from the study after only providing baseline data, two participants withdrew following their first session and two withdrew after their second session. Reasons given for withdrawing (where reported) were feeling too unwell, attending many hospital appointments, caring for a relative who had become ill and making a phased return to work which took up energy. Brief written feedback indicated some barriers identified by three participants: ‘Not having the right footwear so getting wet feet didn't help! Juggling with Zoom, feeding-back, using my phone with gloves etc was a distraction’; ‘My mother was ill and hospitalised during the study and I was able to attend one session only’.

Discussion: In terms of feasibility, the study had a moderate rate of withdrawal given the health challenges faced by individuals struggling with Long-COVID. Following consent and provision of waitlist control period data, 27% did not provide post-intervention data. Only four of these participants withdrew from the study following attendance of 1-2 Forest bathing sessions. Reasons for withdrawal or non-attendance at sessions were: feeling too ill, having medical appointments, or having carer responsibilities. On average, remaining participants attended three Forest bathing sessions. There are few published intervention studies focusing on Long-COVID with which to compare our retention rate with, and the few studies found did not report retention or attrition rates. Once exception was Estebanez-Pérez et al, (2022) who had already allowed for a 40% non-adherence rate (i.e. adherence was classed as Long-COVID patients attending >12 out of 20 sessions of digital physiotherapy) and so were able to achieve full adherence.

However, the manuscript has been written in an impressive manner and I appreciate authors for their work and recommend its Acceptance if addressed well with reviewers’ comments and/or suggestions.

Thank you for this helpful review which has improved the quality of the paper. We hope you will be happy with the changes.

Reviewer 2 Report

This is an interesting paper suggesting the potential of online forest bathing as a treatment for long COVID and I enjoyed reading it. However, there are a few concerns.

  The problems are as follows.
1. It is revealed that various symptoms of Long COVID reduced over time. In your study, you concluded the efficacy of forest bathing.  However, it is difficult to show efficacy because your study did not set the control participants. If you want to mention the efficacy of forest bathing, you should describe the additional data which show the comparison of the intervention group and control group (non-intervention).     2.  Please describe the differences between the forest bathing in your study and that in previous studies.   You mentioned, "The percentage improvements seen in the current study, exceed those seen from in-person Forest bathing sessions which were open to the public (McEwan et al. 2021), the large increase in social connection in this study is particularly striking."     3. You should describe all of your a Long- COVID symptom survey items (15 items scored 1-7) as a SUPPLEMENT file.   You mentioned, " In the absence of a concise measure of Long-COVID symptoms, the authors took the symptoms list from the WHO (World Health Organization) website (extreme tiredness; shortness of breath; chest pain or tightness; problems with memory and concentration; difficulty sleeping; heart palpitations; dizziness; pins and needles; joint pain; depression and anxiety; tinnitus, earaches; feeling sick, diarrhea, stomach aches, loss of appetite; a high temperature, cough, headaches, sore throat, changes to sense of smell or taste; rashes) and created a Long- COVID symptom survey (15 items scored 1-7).  "

Author Response

Reviewer 2

This is an interesting paper suggesting the potential of online forest bathing as a treatment for long COVID and I enjoyed reading it. However, there are a few concerns.

  The problems are as follows.
1. It is revealed that various symptoms of Long COVID reduced over time. In your study, you concluded the efficacy of forest bathing.  However, it is difficult to show efficacy because your study did not set the control participants. If you want to mention the efficacy of forest bathing, you should describe the additional data which show the comparison of the intervention group and control group (non-intervention).  

Authors response: There was a waitlist control group in this study and the data are shown in the tables and figures. We have been through the manuscript and highlighted this better. We have also added our rationale for the choice of control group in the methods section as follows:

The evaluation used a waitlist controlled, mixed-methods repeated measures design., where A waitlist control was used because although the research team wanted to include a control group, discussions with team members who have lived-experience of Long-COVID led to the conclusion that it would not be ethical to monitor this heavily surveyed population who are rarely offered treatment, without offering an active intervention following the monitoring period.

  1. Please describe the differences between the forest bathing in your study and that in previous studies.   You mentioned, "The percentage improvements seen in the current study, exceed those seen from in-person Forest bathing sessions which were open to the public (McEwan et al. 2021), the large increase in social connection in this study is particularly striking."  

Authors response: We have removed this sentence. It is likely the increased percentages are merely due to the current study taking place during a pandemic where anxiety scores were elevated and due to the severity of anxiety and social isolation in this population (i.e. if you start off more ill then you have more room to improve).

  1. You should describe all of your a Long- COVID symptom survey items (15 items scored 1-7) as a SUPPLEMENT file.   You mentioned, " In the absence of a concise measure of Long-COVID symptoms, the authors took the symptoms list from the WHO (World Health Organization) website (extreme tiredness; shortness of breath; chest pain or tightness; problems with memory and concentration; difficulty sleeping; heart palpitations; dizziness; pins and needles; joint pain; depression and anxiety; tinnitus, earaches; feeling sick, diarrhea, stomach aches, loss of appetite; a high temperature, cough, headaches, sore throat, changes to sense of smell or taste; rashes) and created a Long- COVID symptom survey (15 items scored 1-7).  "

Author response: We have included a copy of the scale in supplementary materials and noted this in the methods.

Thank you for this helpful review which has improved the quality of the paper. We hope you will be happy with the changes.

Reviewer 3 Report

 I think that this manuscript is a very meaningful pilot study that presents an alternative for long-term COVID-19 patients. My comments can be taken as suggestions.

1.       This study appears to be a single-group, within-subject, repeated measures design. Present the research design as a figure for the reader's understanding.

2.       The recruitment process for participants is described in a complicated way. It would be good if the recruitment process including drop-out should also be presented as a picture.

3.       The COVID symptom questionnaire was created and used in this study. It would be nice if the tool validation process could be explained.

4.       Describe the demographic characteristics of participants.

5.       In the Discussion section, I propose to add a review of previous studies in people with Long-COVID symptoms.

6.       The discussion is too condensed, so you need to write a more in-depth discussion.

Author Response

Reviewer 3

 I think that this manuscript is a very meaningful pilot study that presents an alternative for long-term COVID-19 patients. My comments can be taken as suggestions.

  1. This study appears to be a single-group, within-subject, repeated measures design. Present the research design as a figure for the reader's understanding.

Authors response: We have included a figure of the procedure in the methods section

  1. The recruitment process for participants is described in a complicated way. It would be good if the recruitment process including drop-out should also be presented as a picture.

Authors response: We have moved all information on drop-outs/retention to the results section and removed any repetition in the paper. We have also provided retention numbers in the figure.

  1. The COVID symptom questionnaire was created and used in this study. It would be nice if the tool validation process could be explained.

Author response: We have included a description of reliability analysis for the scale under the survey subheading of the methods section as follows:

The reliability of this scale was tested using Cronbach’s alpha and found to be very reliable (α=.86), only the removal of the item concerning loss of smell and taste (which showed the lowest average score) might contribute to a greater Cronbach’s alpha score (α=.87).

  1. Describe the demographic characteristics of participants.

Author response: Demographics are described under the participant subheading of the methods section.

  1. In the Discussion section, I propose to add a review of previous studies in people with Long-COVID symptoms.

Authors response: We have extensively added to the previous literature regarding interventions for Long-Covid patients in the Introduction and Discussion sections. This was challenging because although it appears there is previous literature, on closer inspection many search results were actually letters to editors, workshop presentations or protocols for as yet unpublished trials. For example, the following has been added to the discussion:

Intervention studies for Long-COVID are still in development (Hawke et al, 2022) and the few published intervention studies have so far focused on physical rehabilitation to improve breathing, muscle strength and functional capacity (Prabawa et al, 2022; Estebanez-Pérez et al 2022), oxygen therapy to improve fatigue and global cognition (Robins et al 2021), and aromatherapy to improve fatigue (Hawkins, 2022).  None of these intervention studies have focused on exposure to nature or active engagement with nature through Forest bathing. They have also not focused on outcomes such as anxiety, social connection or self-reported symptoms. In these respects, the current study offers a unique insight into an intervention which is feasible to adhere to and can improve these outcomes for people with Long-COVID. Offering a wider range of interventions (Hawke et al, 2022) and self-management approaches have been encouraged (Fowler-Davis et al, 2021) by researchers. This study offers promising early evidence for a new type of accessible intervention which can improve health, wellbeing and social outcomes in Long-COVID patients.

In terms of Long-COVID symptoms, the symptoms with the highest average ratings were tiredness, brain-fog, poor sleep quality, depression and anxiety, consistent with (Sampogna et al, 2022). Given that these were clearly salient symptoms and that the largest effect size was for anxiety and large improvements in social connection were reported in surveys and written feedback, future interventions should focus on improvements in fatigue, brain-fog, sleep, anxiety and social connection.

  1. The discussion is too condensed, so you need to write a more in-depth discussion.

Authors response: We have added a more in-depth discussion of the results and added a section on feasibility and retention outcomes.

Thank you for this helpful review which has improved the quality of the paper. We hope you will be happy with the changes.

Round 2

Reviewer 1 Report

Authors addressed the comments well. This paper could be accepted for publication.

Reviewer 2 Report

You showed a waitlist control group and this made the results understandable.

I appreciate your effort and revision.